# COVID-19 Induces Greater NLRP3 Inflammasome Activation in Obese Patients than Other Chronic Illnesses: A Case–Control Study

**DOI:** 10.3390/ijms26041541

**Published:** 2025-02-12

**Authors:** Raíssa Campos D’Amico, Seigo Nagashima, Lucas Baena Carstens, Karina de Guadalupe Bertoldi, Sabrina Mataruco, Júlio Cesar Honório D’Agostini, Elisa Carolina Hlatchuk, Sofia Brunoro da Silva, Lucia de Noronha, Cristina Pellegrino Baena

**Affiliations:** 1Post-Graduate Program in Health Sciences (PPGCS), Pontifícia Universidade Católica do Paraná, Curitiba 80215-901, Paraná, Brazil; rah_damico@hotmail.com (R.C.D.); seigocap@gmail.com (S.N.); lbc017@gmail.com (L.B.C.); karinabertoldi@hotmail.com (K.d.G.B.); cristina.baena@pucpr.br (C.P.B.); 2School of Medicine, Pontifícia Universidade Católica do Paraná, Curitiba 80215-901, Paraná, Brazil; sabrinamataruco@hotmail.com (S.M.); sofiabrunoro@yahoo.com.br (S.B.d.S.); 3School of Medicine, Universidade Federal do Paraná, Curitiba 80060-240, Paraná, Brazil; juliocesar201483@gmail.com (J.C.H.D.); elisahlatchuk@gmail.com (E.C.H.)

**Keywords:** COVID-19, SARS-CoV-2, inflammasome, pyroptosis, obesity

## Abstract

Obesity has been identified as an independent risk factor for severe COVID-19 unfavorable outcomes. Several factors, such as increased ACE2 receptor expression and chronic inflammation, can contribute to this relationship, yet the activation of the NLRP3 inflammasome pathway is also a key element. Our primary goal was to determine whether chronic NLRP3 inflammasome activation in people with obesity is different in critical COVID-19 and in critical chronic conditions. A retrospective analysis was conducted using clinical data and post-mortem lung tissue samples from 14 COVID-19 patients with obesity (group A) and 9 patients with obesity who died from non-COVID-19 causes (group B). Immunohistochemical analysis assessed twelve markers related to the NLRP3 inflammasome pathway. Group A showed a significantly higher expression of ASC (*p* = 0.0387) and CASP-1 (*p* = 0.0142). No significant differences were found for IL-8, TNF-α, NF-kB, NLRP3, IL-1β, and gasdermin-D. Group B had higher levels of IL-6 (*p* < 0.0001), IL-18 (*p* = 0.002), CASP-9 (*p* < 0.0001), and HIF (*p* = 0.0327). We concluded that COVID-19 activates the NLRP3 inflammasome pathway, possibly leading to pyroptotic cell death mediated by caspase-1. In contrast, people with obesity without COVID-19, despite exhibiting some markers of the NLRP3 inflammasome, are more likely to experience necroptosis mediated by caspase-9.

## 1. Introduction

As we continue to confront the enduring challenges posed by COVID-19 and the persistent circulation of the SARS-CoV-2 virus, critical inquiries into its pathophysiology have gained prominence [1,2]. Emerging evidence has underscored obesity as an independent risk factor for severe disease outcomes [3,4,5]; with obesity rates nearly tripling over the past 40 years, affecting over 650 million people globally, urgent research is needed to understand and address this intersection between obesity and COVID-19 outcomes [6,7]. However, the link between viral infections and obesity is not new, as previous experiences, such as the 2009 H1N1 influenza pandemic, revealed that populations with obesity are at a higher risk of severe complications [8,9].

While the exact mechanisms are still under investigation, some factors contributing to this association include the increased expression of the ACE2 receptor in individuals with obesity [10,11], anatomical characteristics of obesity that affect lung function [12,13], chronic inflammation marked by elevated cytokine levels [14,15], and alterations in lipid metabolism that support viral replication [16]. Obesity fosters a state of chronic low-grade inflammation through adipose tissue dysfunction, characterized by the excessive secretion of pro-inflammatory cytokines such as IL-6, TNF-α, and IL-1β. Additionally, adipose tissue serves as a reservoir for the ACE2 receptor, potentially increasing viral load and facilitating the progression of the infection. This primed inflammatory environment not only predisposes individuals to hyperinflammatory responses, including the cytokine storm, but also exacerbates complications such as acute respiratory distress syndrome (ARDS) [17,18].

The activation of the NLRP3 inflammasome is a critical link between chronic obesity-related inflammation and the exacerbated immune response to SARS-CoV-2. This multiprotein complex, activated by pathogen- and damage-associated molecular patterns (PAMPs and DAMPs), plays a central role in innate immunity [19,20]. In obesity, chronic stimuli like free fatty acids and oxidized lipoproteins maintain NLRP3 activation, increasing IL-1β and IL-18 levels. Acute insults, such as SARS-CoV-2 infection, amplify this activation via ACE2 receptor interaction and pyroptosis induction, intensifying inflammation and tissue damage [21,22].

NLRP3 activation occurs in two phases: priming and activation. During priming, signaling via Toll-like receptors (TLRs) and NF-κB upregulates NLRP3 (NALP) and pro-inflammatory cytokines (pro-IL-1β and pro-IL-18), leaving NALP in an inactive state. The activation phase, triggered by intracellular stress signals (e.g., potassium efflux, mitochondrial dysfunction, and ROS), promotes NLRP3 oligomerization, the recruitment of ASC, and the activation of CASP-1. CASP-1 processes pro-IL-1β and pro-IL-18 into active forms and induces pyroptosis via gasdermin-D cleavage [23].

This study, based on post-mortem lung biopsies of obese individuals who died from COVID-19 or other chronic conditions, aims to evaluate the activation of the NLRP3 inflammasome pathway. The focus is on assessing the expression of key proteins, particularly NLRP3, ASC, and CASP-1. Given that SARS-CoV-2 amplifies NLRP3 activation, we hypothesize that obese patients with COVID-19 will exhibit greater NLRP3 inflammasome activation compared to obese patients without COVID-19.

## 2. Results

Group A comprised 14 patients, predominantly males (11 males, 78.6%). The median age in this group was 68 years (IQR 19.5). The median duration of mechanical ventilation before decease was 15.5 days (IQR 10.5). The most prevalent pre-existing comorbidities in this sample were systemic arterial hypertension (85.71%, *n* = 12), type 2 diabetes mellitus (35.71%, *n* = 5), and dyslipidemia (28.57%, *n* = 4).

Group B consisted of nine participants, also predominantly male (6 males, 66.6%). The median age in this group was 65 years (IQR 23). The median duration of mechanical ventilation before decease was 1 day (IQR 1). The most prevalent pre-existing comorbidities were systemic arterial hypertension (55.55%, *n* = 5), hepatic cirrhosis (33.33%, *n* = 3), and chronic obstructive pulmonary disease (22.22%, *n* = 2). Causes of death in group B varied and included thromboembolic events, metastatic neoplasms, and acute myocardial infarction.

Table 1 presents the descriptive analysis of the clinical and epidemiological characteristics of the populations in groups A and B. Detailed information on the population characteristics is provided in Appendix A.

Regarding the immunohistochemical analysis, there was a statistically significant difference in the expression of the following markers, with higher concentrations in control group B compared to COVID-19 group A: IL-6 (*p* < 0.0001), IL-18 (*p* = 0.002), CASP-9 (*p* < 0.0001), and HIF (*p* = 0.0327). The graphical representation of these results and the immunohistochemistry images are available in Figure 1.

A statistically significant difference in marker expression in favor of COVID-19 group A was found for ASC (*p* = 0.0387) and CASP-1 (*p* = 0.0142). These data can be seen in Figure 2.

No significant difference was found between the two groups in the analysis of the following molecules: IL-8 (*p* = 0.8291), TNF-α (*p* = 0.2349), NF-kB (*p* = 0.9784), NLRP3 (*p* > 0.9999), IL-1β (*p* = 0.5856), and gasdermin-D (*p* = 0.6883). The corresponding graphical representations and immunohistochemistry images can be found in Figure 3.

The statistical analysis of the immunomarkers is summarized in Table 2, providing an overview of the key findings.

## 3. Discussion

Our study aimed to elucidate the intricate involvement of the NLRP3 inflammasome pathway in the context of both obesity and COVID-19. We found that individuals with obesity and COVID-19 exhibited heightened activation of part of the NLRP3 inflammasome pathway, as evidenced by elevated levels of key protein structures such as ASC and CASP-1. This heightened activation probably led to a predominant mechanism of cellular death known as pyroptosis, primarily mediated by the enzyme caspase-1 [24,25]. In contrast, the control group showed a greater immunoexpression of the enzyme caspase-9, suggesting that the primary mode of cellular demise observed in this cohort was conventional apoptosis, a non-inflammatory form of cellular death, mediated by this enzyme [26]. CASP-9 is a key initiator of the intrinsic apoptotic pathway, activated by mitochondrial stress and cytochrome c release. It forms the apoptosome complex with APAF-1, triggering executioner caspases (caspase-3 and -7) and leading to controlled, non-inflammatory cell death. Unlike caspase-1-driven pyroptosis, caspase-9-mediated apoptosis is immunologically silent and essential for tissue homeostasis. Its elevated levels in the control group, combined with the lower expression of caspase-1, further support our hypothesis of greater inflammasome activation in obese individuals with COVID-19 [27,28]. Although key mediators of pyroptosis, such as ASC and CASP-1, were elevated in group A, we have not directly measured pyroptotic cell death and therefore cannot conclusively assert that it is the dominant pathway in these cases. We can only infer the presence of pyroptosis based on the elevated levels of these markers and what is already known from previously studied and published cellular pathways. Therefore, while the data suggest the involvement of the pyroptosis pathway, further studies are needed to definitively prove its presence in this context.

Our study findings also demonstrate no significant differences in the NF-kB marker and the NOD-like receptor NLRP3 between the two groups investigated, indicating potential nuances in the inflammasome activation pathway. The proposed two-phase model for inflammasome activation, involving priming and activation stages, elucidates a complex regulatory mechanism [29,30,31]. Priming involves the transcriptional induction of NLRP3 initiated by NF-kB pathway activation, leading to the generation of inactive NLRP3 proteins. Conversely, full activation requires intracellular alterations such as changes in cytoplasmic K+ and Ca^2+^ levels, facilitating conformational modifications through ubiquitination and phosphorylation. This enables NLRP3 interaction with ASC, forming an active complex recruiting pro-caspase-1 into its active form, CASP-1 [32,33,34]. Recent findings indicate that ASC is not directly involved in the priming process and ASC overexpression alone does not bypass the priming requirement for NLRP3 inflammasome activation. Instead, the ubiquitination and oligomerization of ASC play critical roles in the activation phase, mediated by proteins such as Peli1, HOIL-1L, and TRAF3, which facilitate ASC aggregation and the subsequent recruitment of CASP-1 [35]. Patients in group A likely exhibit full activation of the NLRP3 inflammasome pathway due to constant inflammatory stimuli associated with active SARS-CoV-2 infection. In contrast, group B shows a lower proportion of fully activated NLRP3 inflammasome complexes and a higher proportion of complexes in the priming phase, probably due to lacking the inflammatory stimulus from the viral infection. While the previous literature has explored NLRP3 inflammasome markers in COVID-19 patients, there is a gap in understanding these markers within the populations with obesity, necessitating further investigation for direct comparative analyses [36].

In addition, secondary findings underscored the nonspecific nature of pro-inflammatory cytokine elevation, with no notable differences observed in TNF-α and IL-8 levels between the two groups and higher levels of IL-6 in the control group. This suggests that the inflammatory response provoked by COVID-19 shares commonalities with other severe chronic diseases, highlighting the complexity of cytokine signaling mechanisms. All these cytokines are well known for their significant roles in acute infections and inflammatory responses, as evidenced by conditions such as COVID-19 [37,38]. However, their elevation is not specific to infectious diseases, as they can also be elevated in various other medical conditions, including cardiovascular diseases, atherosclerosis, neoplasms, and autoimmune disorders [39,40,41].

Our study resonates with a previous study [42] that examined cytokine values across various medical conditions, including moderate and severe COVID-19, ARDS, and sepsis, which revealed no significant differences in the concentrations of key cytokines, such as IL-6, IL-8, and TNF-α. These findings suggest that the cytokine levels observed in COVID-19 may not significantly differ from those seen in other severe diseases at a critical course of life. Unamuno et al. [43] assessed the mRNA expression of NLRP3 inflammasome markers in participants with and without obesity. This study revealed elevated IL-6 and TNF-α mRNA levels in the group with obesity compared to the one without, indicating that obesity alone may contribute to increased marker levels. This observation is consistent with our results, as both study groups comprised people with obesity, and it is corroborated by the existing literature, which demonstrates that obesity is associated with chronic inflammation and the persistent elevation of TNF-α and IL-6 levels [44,45]. Furthermore, it is important to note that the methodology used for determining cytokine levels differed, with their study utilizing messenger RNA analysis while ours employed immunohistochemical expression analysis.

Also, it is important to highlight that IL-6 is a well-established acute-phase cytokine that rises rapidly in the early stages of inflammation and subsequently declines as the inflammatory response resolves. The higher IL-6 concentration observed in the control group may be partially explained by the difference in hospitalization duration between the groups. Patients in group B had a shorter length of stay, whereas COVID-19 patients experienced prolonged hospitalization, which could have influenced IL-6 dynamics [46,47].

Following that, also as a secondary outcome, we analyzed IL-1β and IL-18, which are the key cytokines associated with NLRP3 inflammasome activation [48]. IL-1β concentrations did not significantly differ between groups, possibly due to its close association with obesity-related inflammation [49,50]. The literature has already demonstrated the elevated mRNA expression of caspase-1 and IL-1β in patients with obesity and diabetes compared to eutrophic patients. Additionally, it was identified that weight loss significantly reduces the activation of CASP-1 and IL-1β genes [51].

Conversely, IL-18 levels were higher in participants from the control group. Pathways beyond NLRP3 inflammasome activation may contribute to IL-18 production, which has also been directly linked to obesity in previous studies [52]. For instance, Vandanmagsar et al. [53] have shown considerably higher plasma concentrations of this cytokine in mice with obesity fed high-fat diets compared to eutrophic mice. Another study demonstrated elevated plasma IL-18 in individuals with obesity with glycemic disorders compared to a group of healthy eutrophic patients, identifying a positive relationship between increased BMI and serum IL-18 concentration [54]. These findings highlight, again, the nonspecific nature of cytokines, whose production is influenced not only by acute infectious processes but also chronic diseases, including obesity. A recent study reported a significant reduction in mortality rates among obese patients with COVID-19 who were treated with semaglutide. These findings translate the critical impact of obesity on COVID-19 outcomes and offer further evidence of the importance of addressing obesity as a major risk factor in the context of COVID-19 [55].

IL-18 elevation in the control group may also be attributed to its activation through protease-mediated cleavage independent of the NLRP3 inflammasome. Several proteases, including proteinase 3, mast cell chymase, and granzyme B, have been shown to process pro-IL-18 into its biologically active form without requiring caspase-1 activation. These alternative pathways may have contributed to the elevated IL-18 levels observed in the control group, independent of NLRP3 inflammasome activation or pyroptotic signaling [56].

The observed disparities in NLRP3 inflammasome activation and levels of cytokine expression between our studied groups may stem from a multitude of factors, including the presence of comorbidities and the duration of underlying illnesses. Our study is not free of limitations. The modest sample size and logistical hurdles in participant recruitment during a pandemic constrain the generalizability of our findings. We also have to acknowledge that while numerical BMI data were available for all patients in group A, BMI values were not present for group B. The patients in this group were classified as obese throughout categorical anthropometric classifications provided in the autopsy reports. Although these classifications were standardized and rigorously applied, the absence of precise numerical data may introduce variability when comparing the two groups.

Also, we have to take into consideration that the median duration of mechanical ventilation was significantly shorter in the control group compared to group A, and that this difference could potentially influence NLRP3 inflammasome activation levels, as prolonged mechanical stretch is known to exacerbate inflammatory pathways. However, it is also important to note that the existing literature supports the notion that the activation of the NLRP3 inflammasome can occur relatively early during mechanical ventilation, since studies have demonstrated that the NLRP3 pathway can be triggered within a few hours of exposure to mechanical stretch (2–6 h) [57,58,59]. This early induction can lead to the release of pro-inflammatory cytokines such as IL-1β, even in the initial hours of mechanical ventilation. Thus, the shorter duration of mechanical ventilation in group B does not preclude the possibility of NLRP3 activation.

In addition to mechanical stretch, hypoxia may have differentiated the groups and limited our results. Hypoxia independently triggers NLRP3 inflammasome activation via hypoxia-inducible factor (HIF), a heterodimeric transcription factor [60]. The alpha subunit (HIF-1α and HIF-2α) is regulated by tissue oxygen levels, while the beta subunit (HIF-1β) has constitutive expression. Elevated HIF-1α can activate the NF-kB pathway, leading to inflammasome complex production [61]. This activation has been noted in several diseases, suggesting that HIF-1α might modulate inflammasome activation [62].

COVID-19 is widely recognized for causing significant respiratory impairment and acute tissue hypoxia due to gas exchange difficulties in inflamed alveoli [63]. As a result, HIF may be intensely activated in group A, potentially triggering the NLRP3 inflammasome as a consequence of hypoxia rather than directly due to COVID-19. However, our analysis revealed that the control group also exhibited activation of this pathway, likely due to other underlying disease mechanisms that contributed to mortality in this group [64]. Chronic diseases such as hepatic cirrhosis, thromboembolic events, and metastatic neoplasms, which were prevalent in group B, are well-documented causes of systemic hypoxia and upregulation of HIF-1α. Hepatic cirrhosis, for instance, is associated with hypoxic microenvironments due to impaired hepatic oxygen delivery and vascular remodeling, leading to increased HIF-1α activation [65,66]. Thromboembolic events, another cause of mortality in the control group, induce localized and systemic hypoxia, activating HIF pathways that exacerbate inflammation and coagulation [67] Additionally, metastatic neoplasms create hypoxic tumor microenvironments, which drive HIF-1α expression to promote angiogenesis and inflammatory signaling [68]. This finding suggests that hypoxia was not exclusive to the COVID-19 group; it was also observed in the control group, indicating that it likely did not limit our study.

Promising implications arise from our study, which aims to elucidate the intricate involvement of the NLRP3 inflammasome pathway in the context of both obesity and COVID-19. Obesity is characterized by chronic pro-inflammatory conditions, with the NLRP3 inflammasome pathway being closely linked to this process. Our study found that COVID-19 induces inflammation comparable to other severe chronic diseases, as evidenced by similar levels of pro-inflammatory cytokines, and in some cases, even higher levels in control subjects. This underscores the nonspecific nature of cytokine elevation across various pathological mechanisms. However, our findings also suggest that in people with obesity and COVID-19, the predominant inflammatory pathway activated is the NLRP3 inflammasome, probably leading to pyroptotic cell death mediated by caspase-1. Conversely, though individuals with obesity and without COVID-19 exhibit markers of the NLRP3 inflammasome, the main mechanism of cell death is more likely necroptosis, not pyroptosis, mediated by caspase-9, a non-inflammatory form of cell death, non-related to the NLRP3 inflammasome pathway. Moreover, our findings demonstrate that the observed disparities in NLRP3 inflammasome activation between the groups may stem from a multitude of factors, including the presence of comorbidities, especially obesity itself, and the duration of underlying illnesses.

## 4. Materials and Methods

We obtained post-mortem lung tissue samples from two groups: group A (*n* = 14) consisted of patients with obesity who succumbed to COVID-19, while group B (*n* = 9) comprised patients with obesity who passed away due to other non-COVID-19-related diseases. Ethical approval for this study was granted by the National Research Ethics Committee (CONEP) under protocol number 3.944.734/2020, and informed consent was obtained from the families of the deceased.

Samples from group A were collected at Marcelino Champagnat Hospital in Curitiba, Brazil, between 2020 and 2021 through a left anterior mini-thoracotomy. These samples were preserved in 10% formalin solution for subsequent histological and immunohistochemical analyses, and clinical data were obtained by reviewing medical records. Lung tissue fragments from group B were retrieved from the paraffin-embedded tissue sample bank of autopsies conducted at the University Federal of Paraná Clinical Hospital Complex (CHC-UFPR) before the COVID-19 pandemic. Clinical data for this group were extracted from autopsy reports. All the autopsies were performed between 4 and 6 h after the death.

The inclusion criteria for groups A and B were age ≥ 18 years and BMI ≥ 30 kg/m^2^, a defining value for obesity [69]; specifically for group A, there were more inclusion criteria, such as hospital admission due to confirmed COVID-19, as determined by a positive RT-PCR (reverse transcription–polymerase chain reaction) test for SARS-CoV-2 and/or chest tomography suggestive of viral pneumonia; time from death to lung tissue sample collection less than 4 h. The exclusion criteria for groups A and B were participants with other viral diseases, including hepatitis B, hepatitis C, HIV, or other common respiratory viruses and absence of BMI description or anthropometric classification in medical records. It is important to acknowledge that in group A, numerical BMI data were available for all patients, confirming obesity as defined by a BMI > 30 kg/m^2^. In group B, while exact BMI values were not recorded, the autopsy reports provided robust categorical classifications based on standardized anthropometric criteria, ensuring that only patients unequivocally classified as obese were included.

Histological analysis involved staining with hematoxylin and eosin (H&E) and examination under an Olympus BX40 optical microscope (OLYMPUS, Tokyo, Japan). Subsequently, an immunohistochemical assessment was performed for various biomarkers, which are associated with the NLRP3 inflammasome pathway, including interleukin-6 (IL-6), interleukin-8 (IL-8), tumor necrosis factor alpha (TNF-α), nuclear factor-kappa B (NF-kB), NOD-like receptor protein 3 (NLRP3), apoptosis-associated speck-like protein (ASC), caspase-1 (CASP-1), interleukin 18 (IL-18), interleukin-1β (IL-1β), and gasdermin D (GASD-D). We also analyzed caspase-9 (CASP-9), a biomarker of necroptosis, which represents a non-inflammatory death process and hypoxia-inducible factor (HIF), a marker of hypoxia that can trigger the NLRP3 pathway. The immunohistochemical assay followed a protocol where primary antibodies (Appendix A) binding to the target antigen were applied, followed by secondary antibody application (Mouse/Rabbit PolyDetector DAB HRP Brown Detection System, BioSB, Santa Barbara, CA, USA). The technique was revealed using 2,3-diaminobenzidine and hydrogen peroxide substrate to develop a brown color, followed by counterstaining with Harris’s hematoxylin. Specificity controls were performed by (i) omitting the primary antibody (negative control) and (ii) conducting a tissue sample test on positive controls for each immune marker.

Immunomarked slides were scanned using the Axio Scan.Z1 slide scanner (Zeiss, Jena, Germany), and digitized files were divided into high-magnification fields (400×). Twenty to thirty fields of each participant in group A and group B were selected and subsequently analyzed using Image Pro-Plus version 4.5 (Media Cybernetics, Rockville, MD, USA). A semi-automatic measurement of areas with positive immunoreactivity for each marker was conducted and converted into a percentage value for statistical analysis.

We used a modified Allred score to enhance the precision and sensitivity of the analysis by adjusting scores related to the proportion of positive cells and staining intensity under a BX50 optical microscope (OLYMPUS, Tokyo, Japan) and analyzed in ten HPFs (400×, Olympus Objective, 0.26 mm^2^ per sample). The analysis was obtained by summing two scores (proportion and intensity of positivity), ranging from 0 to 8. The proportion score is subdivided according to the percentage of stained cells: score 0: 0% stained cells, score 1: <1%, score 2: 1–10%, score 3: 11–33%, score 4: 34–66%, and score 5: >66%. Meanwhile, the intensity of positivity is evaluated as negative: score 0, weak: score 1, moderate: score 2, and strong: score 3.

Demographic and clinical continuous variables between the two groups (group A and group B) were described using median values and interquartile ranges (IQRs). Data normality was tested using the D’Agostino and Pearson test. Since the data did not follow a normal distribution, non-parametric tests were used. The Mann–Whitney U test was employed to compare clinical variables between the two groups.

For the immunohistochemical analysis, the Mann–Whitney U test was also applied to assess differences in the median values of the percentage of marked area for each immunohistochemical marker across slides from group A and group B. The results were reported as medians and IQRs. Statistical significance was set at *p* < 0.05. Data analysis was conducted using JMP Pro 14.0.0 software (SAS Institute, Cary, NC, USA).

## 5. Conclusions

In conclusion, our study highlights the complexity of inflammatory responses in obesity and COVID-19, particularly related to NLRP3 inflammasome activation. These findings provide valuable insights into understanding the inflammatory mechanisms underlying obesity and its interaction with COVID-19 infection. Furthermore, data gained from studying the NLRP3 inflammasome pathway in the context of COVID-19 may have broader implications for understanding and managing inflammatory responses in other infectious diseases, potentially paving the way for more effective and tailored strategies to combat future pandemics.

## Figures and Tables

**Figure 1 ijms-26-01541-f001:**
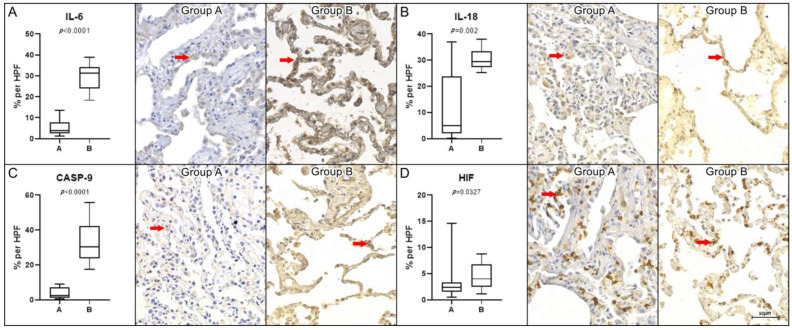
Biomarkers with higher expression in group B. Graphical representation of the percentage of immunoexpression per high-power field (HPF, 400× magnification) for interleukin-6 (IL-6, panel (**A**)), interleukin-18 (IL-18, panel (**B**)), caspase-9 (CASP-9, panel (**C**)), and hypoxia-inducible factor (HIF, panel (**D**)). Statistical analysis revealed significant differences, with lower tissue expression levels (highlighted by red arrows) observed in group A (COVID-19) compared to group B (Control). The corresponding *p*-values were as follows: IL-6 (*p* < 0.0001), IL-18 (*p* = 0.002), CASP-9 (*p* < 0.0001), and HIF (*p* = 0.0327). Statistical significance was determined using the non-parametric Mann–Whitney test (*p* < 0.05). Scale bar = 50 μm.

**Figure 2 ijms-26-01541-f002:**
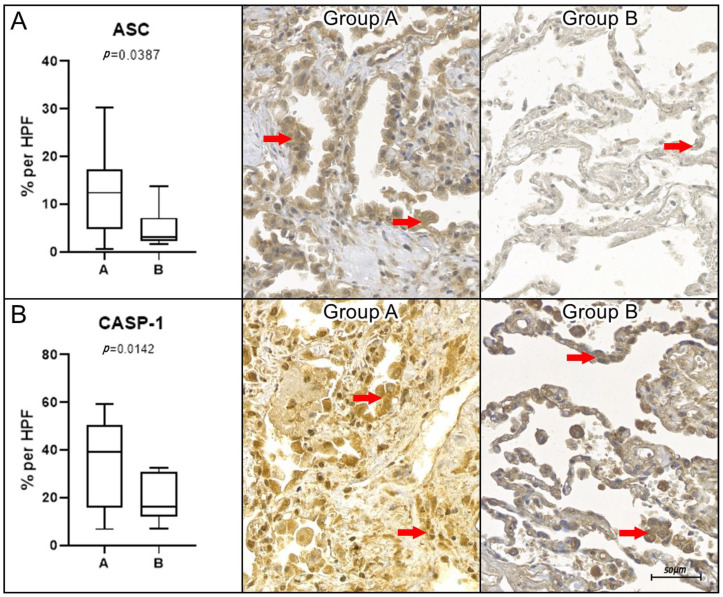
Biomarkers with higher expression in group A. Graphical representation of the percentage of immunoexpression per high-power field (HPF, 400× magnification) for apoptosis-associated speck-like protein (ASC, panel (**A**)) and caspase-1 (CASP-1, panel (**B**)). Statistical analysis indicated significant differences, with higher tissue expression levels (highlighted by red arrows) observed in group A (COVID-19) compared to group B (Control). The corresponding *p*-values were as follows: ASC (*p* = 0.0387) and CASP-1 (*p* = 0.0142). Statistical significance was determined using the non-parametric Mann–Whitney test (*p* < 0.05). Scale bar = 50 μm.

**Figure 3 ijms-26-01541-f003:**
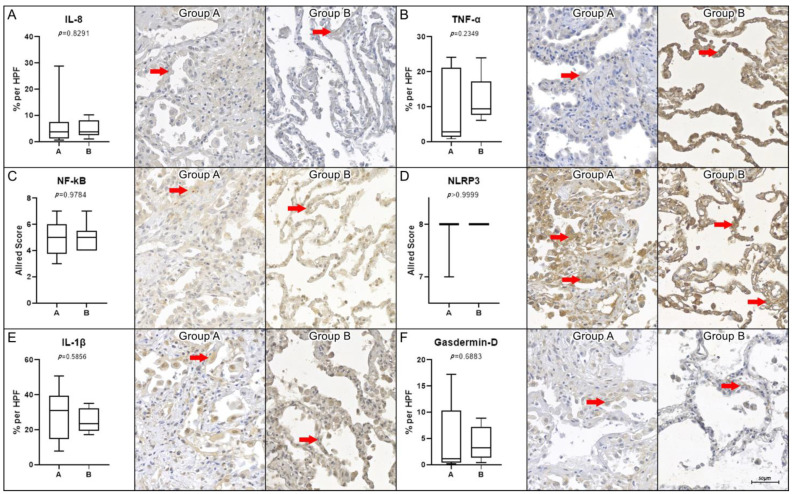
Biomarkers with no statistical difference between the studied groups. Graphical representation of the percentage of immunoexpression per high-power field (HPF, 400× magnification) for interleukin-8 (IL-8, panel (**A**)), tumor necrosis factor alpha (TNF-α, panel (**B**)), nuclear factor-kappa B (NF-κB, panel (**C**)), interleukin-1β (IL-1β, panel (**E**)), and gasdermin D (GASD-D, panel (**F**)). The NOD-like receptor protein 3 (NLRP3, panel (**D**)) is presented using the Allred score per HPF. None of the analyzed markers showed statistically significant differences in tissue expression (highlighted by red arrows) observed in group A (COVID-19) compared to group B (Control). The *p*-values were as follows: IL-8 (*p* = 0.8291), TNF-α (*p* = 0.2349), NF-κB (*p* = 0.9784), NLRP3 (*p* > 0.9999), IL-1β (*p* = 0.5856), and GASD-D (*p* = 0.6883). Statistical analyses were performed using the non-parametric Mann–Whitney test (*p* < 0.05). Scale bar = 50 μm.

**Table 1 ijms-26-01541-t001:** Comparison between groups A and B according to demographic and clinical variables.

Variable		Group A (*n* = 14)	Group B (*n* = 9)
Age (years) ^1^		68.0 (19.5)	65 (23.0)
Sex ^2^	Male	11 (78.6 %)	6 (66.6 %)
	Female	3 (21.4%)	3 (33.4%)
Time from hospitalization to death (days) ^1^		20.5 (13.8)	1 (1)
Mechanical ventilation (days) ^1^		15.5 (10.5)	1 (1)

Variable age expressed in years and time from hospitalization to death and mechanical ventilation expressed in days. ^1^ Median (interquartile range). ^2^ Absolute number (frequency).

**Table 2 ijms-26-01541-t002:** Comparison between groups A and B according to tissue immunoexpression values of biomarkers.

Biomarkers	Group A (*n* = 14)	Group B (*n* = 9)	*p*-Value
ASC	12.4 (10.5)	2.93 (3.53)	0.0387 *
IL-6	3.90 (4.20)	31.1 (9.19)	0.0001 *
IL-18	4.90 (19.0)	29.1 (4.17)	0.002 *
CASP-1	39.2 (31.3)	16.3 (17.5)	0.0142 *
CASP-9	2.47 (5.19)	29.0 (9.06)	0.0001 *
HIF	2.40 (1.26)	4.21 (3.12)	0.0327 *
IL-8	3.70 (5.22)	3.11 (4.90)	0.8291
TNF-α	2.84 (19.4)	8.50 (6.04)	0.2349
IL-1β	30.9 (21.5)	24.1 (10.1)	0.5856
NF-kB	5.00 (2.00)	4.50 (1.25)	0.9784
NLRP3	8.00 (0.00)	8.00 (0.00)	0.9999
Gasdermin-D	1.14 (9.61)	3.55 (4.11)	0.6883

Tissue immunoexpression values of biomarkers expressed in percentual per HPF (high-power field) and characterized as medians. (interquartile range). *p*-values were performed using the non-parametric Mann–Whitney test (*p* < 0.05). Statistically significant differences (*p* < 0.05) are marked with an asterisk (*) in the table.

## Data Availability

The data supporting the findings of this study are publicly available at the following link: https://drive.google.com/drive/folders/1QulxKSkI3C50xARlTTkZXnd8CBosEPVE?usp=sharing (accessed on 24 June 2024).

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
