# Peer review of "COVID-19 Induces Greater NLRP3 Inflammasome Activation in Obese Patients than Other Chronic Illnesses: A Case–Control Study"

_ijms, 2025, doi:10.3390/ijms26041541_

Round 1

Reviewer 1 Report

Comments and Suggestions for Authors

·         The article is interesting and addresses a relevant topic that has affected the world. I suggest some small improvements.

·         Lines 52 to 54 contain a standalone sentence. Combine it with the paragraph to ensure better flow.

·         Your introduction explaining how adipose tissue serves as a reservoir potentially increasing the viral load and facilitating the progression of the infection turned out well.

·         In obesity, NLRP3 inflammasome activation is expected (as explained in line 59). In this study, no difference was observed between the groups, which is consistent given that both groups were obese. This highlights the absence of a non-obese group, which could potentially have shown lower NLRP3 activation compared to the obese groups.

·         As SARS-CoV-2 exacerbates NLRP3 activation, your hypothesis is that Group A would exhibit pronounced NLRP3 activation, as two factors would contribute to this: the condition of obesity itself and SARS-CoV-2. I believe you could include some hypotheses (including ASC e CASP-1) along these lines at the end of the introduction.

·         Since your study is about obesity, I missed data on body mass index (BMI) and body fat to clearly show that the groups were indeed obese.

·         Why is there no "CAUSE OF DEATH" column in Supplementary Table 1, as it is present in Table 2?

·         The panels of the Immunohistochemistry images could be better labeled, including numbers and letters, like in this example: "Immunohistochemistry images showing IL-6 staining in group A (1-left) and group B (2-right).

·         Your focus in the introduction was supported by the NLRP3 inflammasome pathway, but in the absence of differences, you tried to highlight your study in the discussion by emphasizing elevated levels of key protein structures such as ASC and CASP-1. I understand your effort, but it would be important to propose broader hypotheses in the introduction. In fact, ASC and CASP-1 (pyroptotic cell death) were not mentioned at all in the introduction. You need to better articulate what indicates greater NLRP3 activation with other proteins or pathways like ASC and CASP-1, as it seems you stretched the interpretation in the discussion, which in the introduction appeared to be simpler (a pure increase in NLRP3).

·         Some of the increased variables in Group B (IL-6, IL-18, CASP-9, and HIF) could be further explained.

Author Response

 Comments 1:

The article is interesting and addresses a relevant topic that has affected the world. I suggest some small improvements. Lines 52 to 54 contain a standalone sentence. Combine it with the paragraph to ensure better flow. Your introduction explaining how adipose tissue serves as a reservoir potentially increasing the viral load and facilitating the progression of the infection turned out well. In obesity, NLRP3 inflammasome activation is expected (as explained in line 59). In this study, no difference was observed between the groups, which is consistent given that both groups were obese. This highlights the absence of a non-obese group, which could potentially have shown lower NLRP3 activation compared to the obese groups.

Response 1:

Thank you for your thoughtful feedback and suggestions, which have been very important to refine our manuscript. In response to your comments, we have made several adjustments to address your observations.

As requested, we have adjusted the section between lines 52 and 54 to improve the flow of the text. The standalone sentence has been combined with the relevant paragraph, ensuring better coherence. Additionally, we restructured parts of the introduction, particularly to include the hypothesis regarding ASC and CASP-1, highlighting their roles in amplifying NLRP3 inflammasome activation. These changes can be found on page 02, between lines 52 and 74.

We also appreciate your positive feedback regarding the explanation of adipose tissue serving as a reservoir that potentially increases viral load and facilitates the progression of infection. This was a critical point in establishing the relationship between obesity and COVID-19 and framing the context of our study.

Regarding the absence of a non-obese control group, we acknowledge that this comparison could have been insightful. However, obesity is widely recognized as a strong inducer of NLRP3 inflammasome activation. Previous evidence has demonstrated that individuals with obesity exhibit chronic basal activation of the NLRP3 inflammasome, driven by persistent stimuli such as free fatty acids and oxidized lipoproteins, which sustain a low-grade inflammatory state (Bertocchi et al., 2020; López-Reyes et al., 2020; Unamuno et al., 2021; Vandanmagsar et al., 2011).

Considering that the activation of NLRP3 in obesity is well-documented in the literature, we opted to focus our study on comparing the magnitude of this activation in obese patients who died of severe COVID-19 to those with other critical, non-COVID-19 conditions. Our findings revealed that while both groups exhibited activation of the NLRP3 inflammasome, individuals with obesity and COVID-19 demonstrated significantly higher levels of ASC and CASP-1, suggesting a greater activation of this inflammatory pathway and a potential predominance of pyroptotic cell death (You et al., 2022; Frank & Vince, 2019).

We recognize the value of including a non-obese control group in studies exploring NLRP3 activation. However, given the extensive body of evidence already confirming the pro-inflammatory nature of obesity and its role in inducing NLRP3 inflammasome activation, we chose to focus on a clinically relevant question: how the activation of this pathway differs among obese individuals in the presence or absence of severe COVID-19. This approach allowed us to contribute new insights into the interplay between obesity and COVID-19 pathophysiology, without compromising the validity of our findings.

Comments 2:

As SARS-CoV-2 exacerbates NLRP3 activation, your hypothesis is that Group A would exhibit pronounced NLRP3 activation, as two factors would contribute to this: the condition of obesity itself and SARS-CoV-2. I believe you could include some hypotheses (including ASC e CASP-1) along these lines at the end of the introduction.

Response 2:

As requested, we have rewritten significant portions of the introduction to emphasize the role of ASC and CASP-1 in our study. Specifically, we included hypotheses regarding their involvement in amplifying NLRP3 inflammasome activation, particularly in the context of obesity and SARS-CoV-2 infection. These changes highlight the dual contribution of obesity-related chronic inflammation and the acute inflammatory insult caused by SARS-CoV-2, aligning with the hypothesis that Group A would exhibit pronounced NLRP3 activation due to these combined factors. The revised introduction incorporates these elements at its conclusion and can be found in page 02, between lines 52 and 74. We hope these adjustments meet the expectations and add clarity to the manuscript.

Comments 3:

Since your study is about obesity, I missed data on body mass index (BMI) and body fat to clearly show that the groups were indeed obese.

Response 3:

Thank you for your thoughtful observation regarding the inclusion of BMI and body fat data to ensure that the groups in our study were indeed classified as obese.

For Group A, all patients had numerical BMI data documented in their medical records, confirming their classification as obese. However, for Group B, the autopsy reports did not include exact numerical BMI values. Instead, these reports provided categorical classifications of nutritional status, such as cachexia, eutrophic, overweight, and obese. As detailed in the methodology, all autopsies in Group B were conducted prior to the pandemic, and the lung samples were obtained for various studies during that period.

Although we do not have the exact numerical BMI data for Group B, it is important to note that all patients included in the study had complete autopsy reports with adequate anthropometric classifications. Patients classified as cachectic, eutrophic, or overweight were excluded from the study, ensuring that only those classified as obese were included in Group B. This approach ensured consistency in the anthropometric criteria across both groups. To maintain clarity and transparency, we have included this distinction in the methodology and acknowledged it as a limitation of our study in page 9, line 266 to 272 and page 10, between lines 355 and 360.   

Comments 4:

Why is there no "CAUSE OF DEATH" column in Supplementary Table 1, as it is present in Table 2?

Response 4:

Thank you for raising this important point regarding the absence of a "Cause of Death" column in Supplementary Table 1. Group A, whose clinical and epidemiological characteristics are detailed in Supplementary Table 1, consists exclusively of individuals whose cause of death was severe COVID-19 or complications directly related to the infection. As all individuals in Group A shared the same cause of death, we determined that a separate "Cause of Death" column in Supplementary Table 1 was unnecessary. However, we value your observation and hope that this explanation aligns with the study's design and objectives.

Comments 5:

The panels of the Immunohistochemistry images could be better labeled, including numbers and letters, like in this example: "Immunohistochemistry images showing IL-6 staining in group A (1-left) and group B (2-right).

Response 5:

Thank you for your valuable suggestion. As requested, we have revised the figure captions and updated the labeling of the immunohistochemistry images to ensure greater clarity and consistency for the reader. These adjustments aim to make the presentation of results more intuitive and accessible for the audience.

The updated figure captions and image labels can be found on pages 03 and 4 (lines 105-110), 4 (lines 117-121) and 5 (lines 131-138). We believe these changes enhance the clarity of the results and better align with your recommendations.

Comments 6:

Your focus in the introduction was supported by the NLRP3 inflammasome pathway, but in the absence of differences, you tried to highlight your study in the discussion by emphasizing elevated levels of key protein structures such as ASC and CASP-1. I understand your effort, but it would be important to propose broader hypotheses in the introduction. In fact, ASC and CASP-1 (pyroptotic cell death) were not mentioned at all in the introduction. You need to better articulate what indicates greater NLRP3 activation with other proteins or pathways like ASC and CASP-1, as it seems you stretched the interpretation in the discussion, which in the introduction appeared to be simpler (a pure increase in NLRP3).

Response 6:

Thank you for your valuable feedback. We have carefully addressed your comments by revising the introduction to include more detailed information on the role of ASC and CASP-1 in the NLRP3 inflammasome activation pathway. Specifically, we expanded on the two key phases of inflammasome activation — priming and activation — to provide a more comprehensive explanation of how these proteins contribute to the pathway.

Additionally, as per your suggestion, we have included a hypothesis emphasizing the involvement of these key proteins, ASC and CASP-1, in the context of inflammasome activation, particularly under the dual influence of obesity-related chronic inflammation and the acute inflammatory stimulus caused by SARS-CoV-2. These changes aim to create a clearer alignment between the introduction and the discussion, addressing the potential mechanisms underlying the findings.

Furthermore, we have added a new and updated reference (2025) to the discussion section to further support our hypothesis that the differences observed between the two groups are likely due to the patients being in different phases of inflammasome activation. This additional reference provides recent insights into the regulation and dynamics of the NLRP3 pathway, reinforcing our interpretation.

These changes can be found on page 02, between lines 52 and 74 and on page 7, between lines 182 and 187. We hope these adjustments adequately address your recommendations and enhance the clarity, coherence, and scientific support of our manuscript.

Comments 7:

Some of the increased variables in Group B (IL-6, IL-18, CASP-9, and HIF) could be further explained.

Response 7: Thank you for your insightful comment regarding the increased levels of IL-6, IL-18, CASP-9, and HIF in Group B. We have carefully revised the discussion section of our manuscript and added further explanations to better contextualize these findings. Below, we summarize the key points that have been incorporated:

IL-6: We have expanded our discussion to emphasize that IL-6 is a well-established acute-phase cytokine. Given that the length of hospitalization was significantly shorter in Group B, this could partially explain the higher levels of IL-6 observed in this group. Since IL-6 tends to peak in the early stages of inflammation and then decrease over time, the prolonged hospitalization of COVID-19 patients in Group A might have contributed to lower IL-6 levels at the time of sample collection. Added in: Page 8, Lines 224-229.

IL-18: We have highlighted that, in addition to being a cytokine directly linked to obesity, IL-18 has activation pathways beyond the NLRP3 inflammasome. These alternative pathways, such as protease-mediated cleavage, could explain its elevated levels in Group B, independent of inflammasome activation. This reinforces the idea that IL-18 expression is influenced by multiple regulatory mechanisms beyond SARS-CoV-2 infection. Added in: Page 9, Lines 254-260.

CASP-9: We have provided additional details regarding the role of caspase-9 in cell death pathways. The elevated concentration of caspase-9 in Group B, combined with the reduced levels of caspase-1 in this group, further supports our hypothesis that the NLRP3 inflammasome pathway is more active in Group A. This suggests that the predominant cell death mechanism in the control group is likely mediated by caspase-9 rather than caspase-1, reinforcing the idea that inflammasome activation plays a more significant role in COVID-19 cases. Added in: Page 7, Lines 158-164.

HIF: We have expanded our discussion on the baseline conditions of patients in Group B and provided additional evidence supporting the activation of the HIF-1α pathway in these individuals. Several comorbidities present in Group B, such as hepatic cirrhosis, thromboembolic events, and metastatic neoplasms, are known to induce HIF-1α activation. This could explain why HIF levels were markedly elevated in the control group, even when compared to the COVID-19 group, which also had multiple factors contributing to increased HIF expression. Added in: Pages 9 and 10, Lines 300-310.

We hope these additions provide a more comprehensive understanding of the observed differences between groups.

Reviewer 2 Report

Comments and Suggestions for Authors

The authors measured levels of inflammatory cytokines and NLRP 3 in obese patients who died of Covid 19 and obese patients who died of other causes. The mean respiratory times of patients who died of COVID were significantly higher than those who died of other causes. A control group would have been more likely to have died after a similar ventilation time, as prolonged mechanical ventilation is known to have lung-damaging effects.

In the text has been stated:

A statistically significant difference in marker expression in favor of COVID-19 112 group (A) was found for ASC (p=0.0387) and CASP-1 (p=0.0142).

No significant difference was found between the two groups in the analysis of the

126

following molecules: IL-8 (p=0.8291), TNF-α (p=0.2349), NF-kB (p=0.9784), NLRP3

127

(p>0.9999), IL-1β (p=0.5856) and gasdermin-D (p=0.6883)

This is not reflected in the table. Please explain the difference.

Author Response

  1. Point-by-point response to Comments and Suggestions for Authors Comments 1:

The authors measured levels of inflammatory cytokines and NLRP 3 in obese patients who died of Covid 19 and obese patients who died of other causes. The mean respiratory times of patients who died of COVID were significantly higher than those who died of other causes. A control group would have been more likely to have died after a similar ventilation time, as prolonged mechanical ventilation is known to have lung-damaging effects.

Response 1:

Thank you for your comment and the opportunity to clarify our study findings.

We acknowledge that there was a significant difference in mechanical ventilation duration between the groups, which could influence lung inflammation and NLRP3 inflammasome activation. However, it is essential to highlight that both groups were exposed to hypoxia and systemic inflammation, regardless of the underlying cause of death (pages 9–10, lines 275–311).

Our study demonstrated that NLRP3 inflammasome activation is not exclusively dependent on prolonged mechanical ventilation. As documented in the literature, this activation can occur in the early phases of the disease, within just a few hours of exposure to mechanical stress, as reported by Kuipers et al., Grailer et al., and Wu et al. Moreover, NLRP3 activation can also be directly triggered by hypoxia, independent of the underlying etiology, through the hypoxia-inducible factor (HIF-1α) pathway, as demonstrated by AbdelMassih et al., Jahani et al., and Tian et al.

It is important to note that, in our study, immunostaining for HIF-1α was significantly higher in the control group (p=0.0327), indicating that these patients also exhibited a substantial degree of tissue hypoxia despite not having SARS-CoV-2 infection. This finding can be explained by the underlying clinical conditions in the control group, which included cirrhosis, thromboembolic events, and metastatic neoplasms—diseases well-documented to activate HIF-1α. This finding reinforces that, although the COVID-19 group experienced a longer clinical course and extended mechanical ventilation, hypoxia was not an exclusive factor for this group. Thus, our results demonstrate that, despite the difference in ventilation duration, both groups were exposed to hypoxic stimuli capable of inducing NLRP3 activation.

However, only COVID-19 patients exhibited preferential activation of the inflammatory pathway mediated by ASC (p=0.0387) and CASP-1 (p=0.0142), suggesting a predominance of pyroptosis in these cases. In contrast, the control group showed higher CASP-9 activation (p<0.0001), consistent with apoptosis, a non-inflammatory cell death process unrelated to the NLRP3 inflammasome pathway.

Comments 2: A statistically significant difference in marker expression in favor of COVID-19 group (A) was found for ASC (p=0.0387) and CASP-1 (p=0.0142). No significant difference was found between the two groups in the analysis of the following molecules: IL-8 (p=0.8291), TNF-α (p=0.2349), NF-kB (p=0.9784), NLRP3 (p>0.9999), IL-1β (p=0.5856) and gasdermin-D (p=0.6883). This is not reflected in the table. Please explain the difference.

Response 2:

Upon careful review, we have verified that the p-values in Table 2 are consistent with the results described in the manuscript. For example, the statistically significant differences for ASC (p=0.0387, page 4, line 113) and CASP-1 (p=0.0142, page 4, line 113), as well as the non-significant results for IL-8 (p=0.8291, page 4, line 124), TNF-α (p=0.2349, page 4, line 124), NF-kB (p=0.9784, page 4, line 124), NLRP3 (p>0.9999, page 4, line 125), IL-1β (p=0.5856, page 4, line 125), and gasdermin-D (p=0.6883, page 4, line 125), are equally represented in the table and the text. Similarly, other significant findings, such as IL-6 (p=0.0001, page 3, line 100), IL-18 (p=0.002, page 3, line 100), CASP-9 (p=0.0001, page 3, lines 100 and 101), and HIF (p=0.0327, page 3, line 101), are correctly described in the manuscript and in the table.

We understand that the table’s visual presentation might not have clearly differentiated statistically significant values, which could have led to some confusion. To improve clarity and readability, we have revised the table by marking statistically significant values (p < 0.05) with an asterisk (*) and adding a note to explain this formatting. This adjustment should help readers quickly identify the significant findings and ensure consistency between the text and the table.

We appreciate your careful review and constructive feedback, which have allowed us to enhance the clarity of the manuscript.

Round 2

Reviewer 2 Report

Comments and Suggestions for Authors

no comment